# Symmetrical Efficient Gait Planning Based on Constrained Direct Collocation

**DOI:** 10.3390/mi14020417

**Published:** 2023-02-10

**Authors:** Boyang Chen, Xizhe Zang, Yue Zhang, Liang Gao, Yanhe Zhu, Jie Zhao

**Affiliations:** State Key Laboratory of Robotics and System, Harbin Institute of Technology, Harbin 150001, China

**Keywords:** robotics, symmetrical biped gait planning, trajectory optimization, energy efficiency

## Abstract

Biped locomotion provides more mobility and effectiveness compared with other methods. Animals have evolved efficient walking patterns that are pursued by biped robot researchers. Current researchers have observed that symmetry is a critical criterion to achieve efficient natural walking and usually realize symmetrical gait patterns through morphological characteristics using simplified dynamic models or artificial priors of the center of mass (CoM). However, few considerations of symmetry and energy consumption are introduced at the joint level, resulting in inefficient leg motion. In this paper, we propose a full-order biped gait planner in which the symmetry requirement, energy efficiency, and trajectory smoothness can all be involved at the joint level, and CoM motion is automatically determined without any morphological prior. In order to achieve a symmetrical and efficient walking pattern, we first investigated the characteristic of a completely symmetrical gait, and a group of nearly linear slacked constraints was designed for three phases of planning. Then a Constrained Direct Collocation (DIRCON)-based full-order biped gait planner with a weighted cost function for energy consumption and trajectory smoothness is proposed. A dynamic simulation with our newly designed robot model was performed in CoppliaSim to test the planner. Physical comparison experiments on a real robot device finally validated the symmetry characteristic and energy efficiency of the generated gait. In addition, a detailed presentation of the real biped robot is also provided.

## 1. Introduction

Compared with wheeled, crawler, and wriggled locomotion, legged locomotion exhibits significant mobility and effectiveness in complex environments, such as in the case of disasters, hills, and battlefields [1]. Biped robots also have stronger environmental adaptability than multi-legged robots [2,3]. Natural walking patterns, such as bird-like biped walking gait (BBWG) from ostrich and human-like biped walking gait (HBWG), can realize energy-efficient biped locomotion, which is also the pursuit of biped robot researchers [4,5,6,7].

Symmetry is an important feature of natural biped locomotion. From the biological perspective, symmetry has been widely observed in animal biped gaits for stable and efficient locomotion [4,8,9]. Current researchers have also declared that the energy-optimal gaits found in biped robotic research closely resemble the gaits found in nature [5,10,11,12]. Therefore, symmetry has become a key criterion for achieving natural walking patterns for energy efficiency and adaptive locomotion. In this paper, we focus on how to synthesize an energy-efficient symmetrical walking gait.

Different methods have been used to generate a symmetrical efficient complete biped gait consisting of a left support phase (LSP) and a right support phase (RSP). Morphology characteristics are usually directly engaged in generating asymmetrical gait. Kumar et al. [13] introduced approximate symmetry into LSP and RSP through the dynamic similarity principle of gaits (DSPG), but a feed-forward neural network, which was trained using a primitive gait data set, is additionally needed for gait selection. A simplified model with a constrained center of mass (CoM) was also considered. J. Ding et al. [14] first utilized a linear inverted pendulum (LIP) and the CoM-Acceleration-based Optimal Index to synthesize an energetically efficient CoM trajectory for a single phase and then generated joint-level trajectories through inverse kinematics. Although the energy consumption of CoM can be explicitly considered, the joint-level trajectories are still not taken into account. Moreover, feasible dynamic constraints, such as joint input limits and mass distribution of each link, are not introduced either. W. Yu et al. [15] developed a Reinforcement Learning (RL)-based low-energy gait generator with a loss function of symmetrical motions, and the method does not require any morphology knowledge as prior. However, since its total training time is about 4 to 15 h and requires careful parameter tuning, it is difficult to apply to a real robot. Due to various constraints in the biped locomotion process, trajectory optimization (TO)-based methods have inherent advantages when finding feasible trajectories for specific purposes [16,17,18]. Ding J et al. [19] proposed an energy-efficient gait generator via a two-stage optimization w.r.t. CoM trajectory as well as a joint mechanical work, and then a completely symmetrical gait can be realized through the composition of two identical unit walking cycles. Because the method is still derived from LIP, the energy efficiency cannot be fully considered, and a reference zero moment point trajectory is a prerequisite. Felis et al. [20] applied a contact invariant optimization algorithm to generate whole-body locomotion with muscle-based lower-body actuation for humanoids. Although symmetry and periodicity can be explicitly enforced to generate realistic locomotion, an artificial angular momentum cost function must be elaborated to constrain the motion of CoM. Moreover, due to the hypothesis of unlimited joint torques and the engagement of biomechanical data, this method cannot directly apply to a real robot for gait planning.

In order to achieve a feasible, symmetrical, and efficient walking pattern, dynamics and motion details should be sufficiently considered. Constrained Direct Collocation (DIRCON), proposed by Michael Posa et al. [21], can perform full-order dynamics gait planning in relation to the implicitly constrained state manifold caused by kinematic position constraints and their derivatives. Furthermore, pattern constraints are also easy to insert, so it is a good potential framework for generating a dynamic-feasible biped gait with a specific purpose. In this paper, we propose a DIRCON-based full-order biped gait planner whose symmetry requirements, energy efficiency, and trajectory smoothness can all be involved at the joint level, and CoM motion can be automatically determined without any morphology prior. Our method is an end-to-end motion planner wherein no additional technique is needed for assistance, such as the feed-forward neural network for gait selection in Kumar’s work [13]. Moreover, we also avoid the time-consuming and complicated training process in the RL method of W. Yu [15]. Moreover, due to the full-order dynamics, more motion details can be introduced and, unlike in the method proposed by Ding, J [19], where the energy consumption is considered at the CoM level, our method can extend the consideration to the joint level, leading to improved energy efficiency. Therefore, our proposed planner can provide an easier and more convenient way to generate symmetrical and more efficient biped gaits. The contributions of this work can be summarized as follows:

We investigated the symmetric characteristic of a complete gait and designed a group of approximately linear slacked constraints for periodicity and symmetry, which can be used in other optimization-based biped gait planners.

A DIRCON-based planner was proposed taking the symmetry, energy efficiency, and smoothness of the joint trajectories into consideration. A real biped robot is able to walk stably through a feed-forward PD tracking of the generated joint trajectories.

We also delivered a detailed presentation to our newly designed biped robot, which was used to validate our method. The work in this paper preliminarily realizes the intention of developing a biped robot with efficient natural walking.

This paper is organized as follows. In Section 2, we review the theoretical foundations of DIRCON and how to implement multi-phase planning. In Section 3, we first investigate the characteristics of the periodicity and symmetry of a complete gait, and then a group of approximate linear slacked constraints for symmetry and periodicity is formulated. Finally, the DIRCON-based planner is proposed. A detailed presentation of the mechanical design and the onboard components of our newly designed robot is presented in Section 4. In Section 5, we first check the capability of the proposed planner and the robot model through a planar dynamic simulation using CoppeliaSim. Then, physical comparative walking experiments between the proposed planner and the original DIRCON planner are conducted on the real robot. An analysis of the gait morphology, joint behavior, and energy efficiency is finally performed. Section 6 concludes this paper and speculates on future work.

## 2. Related Work

### 2.1. Trajectory Optimization and Constrained Direct Collocation

TO is able to find feasible joint trajectories due to its natural ability to deal with constraints, particularly in high-dimensional constrained state space. Direct Collocation (DC) is a widely used method in biped gait planning, wherein knots are engaged to divide the target trajectories into multiple segments. Through the properties of a Simpson integration, DC inserts the high-dimensional nonlinear dynamics as a collocation constraint at the collocation point, which is usually located at the midpoint between two adjacent knots. Cubic Hermite splines and first-order curves are utilized to interpolate the joint state trajectories and input trajectories, respectively. Thus, DC can avoid the numerical integration present in the Shooting Method [22,23], and the generated trajectories have third-order accuracy, which is the key requirement for stable trajectory tracking [24]. The general form of DC with N+1 knots is shown as follows:(1)$$\underset{z}{\min}\;l_{f}\left(X_{N}\right)\;+\;h_{k}\overset{N-1}{\underset{k=0}{\sum }}l\left(x_{k},u_{k}\right)\;s.t.\;\;\mathrm{for}\;k=0,1,2,\ldots N-1\;g\left(x_{k},u_{k},x_{k+1},u_{k+1}\right)={\;\dot{x}}_{t_{c,k}}\;-\left[\begin{array}{c} v\\ f\left(x_{t_{c,k}},u_{t_{c,k}}\right) \end{array}\;\right]=0\;m\left(z\right)\leq 0\;$$
where z is the set of decision variables consisting of $x_{k}$, $u_{k}$, and $h_{k}$. k represents the time index of the target trajectories, and $h_{k}$ is the time step. $t_{c,k}$ is the midpoint of the time between two knots where the collocation point is usually located. $l_{f}$ and l are the final state cost and process cost, respectively. In other words, the collocation constraint directs the search for optimization to a dynamic feasible local optimal solution considering the current cost function. g(z) is the collocation constraint whose value is decided by the difference of the slope of the Hermite spline and robot dynamics at the collocation point. The relevant constraints, such as joint limits and input limitations, are represented by m(z). More details can be found in [24].

Posa et al. [21] found that the derivatives of the kinematic position constraint will further compress the state space to a constrained manifold, and DIRCON was proposed to carry out robot motion planning in the manifold. The exact form of DIRCON is shown as follows:(2)$$\underset{z}{\min}\;\;l_{f}\left(X_{N}\right)\;+\;h\overset{N-1}{\underset{k=0}{\sum }}l\left(x_{k},u_{k}\right)\;s.t.\;\;\;\;\;\;\bar{g}\left(x_{k},u_{k},x_{k+1},u_{k+1},\lambda_{k},\lambda_{k+1},{\bar{\lambda }}_{k},{\bar{\delta }}_{k}\right)\;=\;{\dot{\bar{x}}}_{t_{c,k}}\;-\left[\begin{array}{c} v+J{\left(q_{t_{c,k}}\right)}^{T}{\bar{\delta }}_{k}\\ \bar{f}\left(x,u,{\bar{\lambda }}_{k}\right) \end{array}\;\right]=0,\;\varphi \left(q\right)=0,\;\;\dot{\varphi }\left(q\right)=\phi \left(q,v\right)=0,\;\ddot{\varphi }\left(q\right)=\psi \left(q,v,\;\dot{v}\right)=0,\;\mathrm{for}\;k=0,1,2,\ldots N-1\;m\left(z\right)\leq 0\;$$
where $\lambda_{k},\;\;{\bar{\lambda }}_{k},\;\;{\bar{\delta }}_{k}$ are the contact forces, force correction, and velocity correction, respectively. ${\dot{\bar{x}}}_{t_{c,k}}$ is the slope of the spline at the collocation point of DIRCON, and the location of the collocation point is set to the midpoint between the adjacent knots, as usual. Furthermore, due to the explicit introduction of contact force λ, some constraints, such as friction limitations, can be easily introduced, which is especially suitable for biped gait planning. As a result, DIRCON provides a potential framework for biped gait planning for specific purposes and maintains the advantages of DC.

### 2.2. Virtual Knots and Multi-Phases Gait Planning

The walking process of a biped can be regarded as a hybrid system consisting of several independent dynamic systems in different phases, and the discrete state transitions are triggered by guard conditions. A complete gait starts at the beginning of LSP and completes at the end of RSP. The state transition occurs when the swing foot contacts the ground. For symmetrical complete gait generation, the discrete transition should be explicitly introduced during planning. The method of virtual knots has been designed in our previous work [18], which we briefly recover here. Assuming the robot is switching from LSP to RSP and that the LSP has $m_{i}$ knots of ${\mathrm{mode}}_{i}$ and that RSP has $m_{i+1}$ knots of ${\mathrm{mode}}_{i+1}$. In order to maintain the trajectory continuity between the two modes, the last knot of ${\mathrm{mode}}_{i}$ should be equal to the first knot of ${\mathrm{mode}}_{i+1}$, i.e., the total sequence takes $m_{i}+m_{i+1}-1$ knots. When contact happens, the position of the first knot of ${\mathrm{mode}}_{i+1}$ is equal to the last knot of ${\mathrm{mode}}_{I}$, and the velocity undergoes a discrete transition that can be modeled by the momentum observer [25]. Then, the virtual knot is established, as seen in Equation (3), and inserted at the beginning of ${\mathrm{mode}}_{i+1}$. As a result, the transition can be characterized as in Equation (4) so that the two adjacent modes are related by the virtual knot constraint, and the trajectories of each phase can be planned independently.
(3)$$\mathrm{Virtual}\;\mathrm{knot}=\left[\begin{array}{c} q_{\mathrm{vir}}\\ v_{\mathrm{vir}} \end{array}\right]=\left[\begin{array}{c} q_{\mathrm{pre}}\\ v_{\mathrm{post}\_i} \end{array}\right]$$
(4)$$J{\left(q_{\mathrm{pre}}\right)}^{T}\mathrm{Impulse}=K_{O}M\left(q_{\mathrm{pre}}\right)\left(v_{\mathrm{post}}-v_{\mathrm{pre}}\right),\;q_{\mathrm{pre}}=q_{\mathrm{post}},$$

Impulse is the Ground Reaction Force (GRF) of the contact point. ${\left[q_{\mathrm{pre}},{\;v}_{\mathrm{pre}}\right]}^{T}\;$and ${\left[q_{\mathrm{post}},{\;v}_{\mathrm{post}}\right]}^{T}$ are the state vector before and after contact, respectively. $K_{O}$ is the diagonal gain matrix of the observer [25], and $J\left(q_{\mathrm{pre}}\right)$ is the Jacobian matrix of the contact point. $M\left(q_{\mathrm{pre}}\right)$ is the inertia matrix before contact. The insertion results in that the whole sequence occupies $\left(m_{i+1}+m_{i}\right)$ knots.

## 3. Symmetrical Efficient Gait Planning

### 3.1. Symmetry Characteristic and Constraints for Three Phases Planning

The symmetry characteristics of a complete gait include phase duration symmetry, foot location symmetry, joint trajectories symmetry, and CoM motion symmetry. Phase duration symmetry means that the duration of LSP and RSP should be equal if the dynamics of each leg are identical. Foot location symmetry means that the step length of each phase is also equal. Joint trajectories symmetry represents an inner constraint at the joint level, which is required by a natural gait pattern. CoM motion symmetry means that the ensemble motion of a bipedal robot in LSP and RSP should behave similarly. The simplified model methods usually engage with phase duration symmetry, foot location symmetry, and CoM motion symmetry, and little consideration is given to the joint level, meaning that the gait is not actually efficient. On the other hand, full-order gait planners are able to adopt all of the symmetry characteristics to generate the gait, and no artificial priors or constraints are needed to design a target motion of CoM. Moreover, due to the inherent error of the mechanical structure, the dynamics of the left and right legs cannot be completely consistent; therefore, we need to leave a certain margin for the planner. In our method, we utilize three symmetries, and the CoM motion is determined automatically by the planner according to the dynamics, and it is able to acquire a higher energy efficiency.

For brevity, we use a seven-link XOZ planar biped robot as an example of a single complete gait plan, as shown in Figure 1. We assumed that the phase sequence includes three modes (${\mathrm{mode}}_{0}$, ${\mathrm{mode}}_{1}$, and ${\mathrm{mode}}_{2}$) with m knots, n knots, and p knots, respectively. It is worth noting that the three phases together constitute one complete gait, i.e., ${\mathrm{mode}}_{0}$ and ${\mathrm{mode}}_{2}$ comprise the LSP, and ${\mathrm{mode}}_{1}$ represents RSP.

Now the symmetry characteristic and periodicity can be introduced via a group of approximate linear constraints. We utilize three decision variables $x_{0}$, $x_{1}$, and $x_{2}$ to represent the locations of one complete gait, as shown in Figure 1. A premier feature of foot location symmetry is that $x_{1}$ should be the midpoint of $x_{0}$ and $x_{2}$. To reduce the difficulty of solving the optimization, we release the mid location by using a slack variable, and the constraint can be described as seen in Equation (5):(5)$$x_{1}=\left(0.5+{\mathrm{slack}}_{0}\right)(x_{0}+x_{2}),\;a\leq {\mathrm{slack}}_{0}\leq b$$
where ${\mathrm{slack}}_{0}$ is the slack variable and can be set to a small decimal near zero to scale the mid footholds. a and b are the boundaries of ${\mathrm{slack}}_{0}$ and can be set manually between [−0.5,0.5] according to some additional requirements, such as terrain information or location preference. Furthermore, 0.5 means that the robot would step forward from the LSP, and RSP marches one spot; −0.5 means the opposite. If ${\mathrm{slack}}_{0}$ is forced to zero, the step length of the LSP and RSP are strictly restricted to be equal. In this paper, due to the mechanical error of the left and right legs, we set a and b to −0.05 and +0.05 to obtain a higher solution success rate. A weighted cost function of ${\mathrm{slack}}_{0}$ is engaged to induce $x_{1}$, approaching the midpoint. Although the constraint ensures the symmetry of foot location, it is not sufficient to lead to the formation of a symmetrical gait pattern.

As for the symmetry of the duration and joint trajectories, we explicitly constrain the duration of the LSP and RSP to be equal, and two linear constraints are formulated, which are that when the swing foot contacts the ground, the joint states should be exchanged between the legs to introduce the symmetry into joint-level trajectories. One may consider that symmetry can be directly inserted into joint trajectories via equation constraints on each knot. However, it is too strict for the solver and very redundant. Moreover, there still exist some mechanical errors between the left and right legs, so strict kinematic constraints are not appropriate for solving optimization. Instead, by using linear state exchange constraints, the intermediate joint trajectories can be determined automatically by the solver, and the generated trajectories also behave approximately symmetrical. Furthermore, for periodicity, the final state of ${\mathrm{mode}}_{2}$ should be equal to the initial state of ${\mathrm{mode}}_{0}$. As a result, a group of linear constraints for the duration and joint trajectories symmetry is formulated, as seen in Equation (6), including a duration constraint, two state exchange constraints, and an equality of the initial state and final state. $h_{t}$ is the time step of each segment, and H is the total duration of a single complete gait w.r.t. gait speed. $x_{\mathrm{lf}\_m_{0}}$, $x_{\mathrm{rf}\_m_{0}}$, $x_{\mathrm{lf}\_m_{1}}$, and $x_{\mathrm{rf}\_m_{1}}$ are the joint states of the left and right legs at the end of ${\mathrm{mode}}_{0}$ and ${\mathrm{mode}}_{1}$, respectively.
(6)$$\overset{n-1}{\underset{t=0}{\sum }}h_{t}=\overset{m-1}{\underset{t=0}{\sum }}h_{t}+\overset{p-1}{\underset{t=0}{\sum }}h_{t}=0.5*H\;x_{\mathrm{lf}\_m_{0}}=x_{\mathrm{rf}\_m_{1}}\;x_{\mathrm{rf}\_m_{0}}=x_{\mathrm{lf}\_m_{1}}\;\mathrm{Initial}\;\mathrm{state}=\mathrm{Final}\;\mathrm{state}$$

One may consider the reason why we utilize three-phase planning to generate a complete symmetrical gait when two-phase planning is enough to achieve the desired gait. It seems to be convenient to form a symmetrical gait with two phases, but this is not actually the case. From the perspective of duration symmetry, the duration constraint of two-phase planning can be directly expressed as the duration of ${\mathrm{mode}}_{0}$ being equal to ${\mathrm{mode}}_{1}$, and the constraint of three-phase planning also only requires that the total duration of ${\mathrm{mode}}_{0}$ and ${\mathrm{mode}}_{2}$ is equal to ${\mathrm{mode}}_{1}$. Since both of them are linear, they are equivalent to the solver. Similarly, some other linear constraints, as shown in Equations (5) and (6), of two-phase planning are also equivalent to three-phase planning. However, the situation is different for nonlinear constraints, such as swing foot motion. Two-phase planning requires the swing foot to come in contact with the ground at the beginning and end of the gait, while three-phase planning does not. Two methods can be considered to maintain this contact. First, one may try to find the exact initial state and the final state that satisfy the contact constraint. However, it is difficult to obtain an appropriate position because infinite solutions exist if the CoM position is not constrained, and other techniques should be engaged to find the approximate state of CoM. Second, one may utilize a periodic contact constraint without any constraint of CoM. However, the nonlinear contact constraints of the swing foot still exist. Furthermore, some related constraints, such as friction limitations, are also needed, especially when the contact is in a multi-point model. The other nonlinear constraints are equivalent to two-phase plans. Therefore, our three-phase planning is an easier and more convenient method than two-phase planning for symmetrical gait generation.

### 3.2. DIRCON-Based Symmetrical Efficient Gait Planner

Due to the energy efficiency at the joint level and the trajectory smoothness, we introduce them with two highly weighted cost functions, as shown in Equation (7). Finally, with the combination of Equations (6) and (7), and the weighted cost function, the planner is able to generate a symmetrical efficient natural gait using three-phase planning. The exact formulation of the proposed planner can be written as follows:(7)$$\underset{z}{\min}\;l_{0}\left(x_{0}\right)+l_{f}\left(x_{N}\right)+K_{0}h_{m}\sum_{k=1}^{N}l\left(x_{k},u_{k}\right)+K_{1}{\mathrm{slack}}_{0}{}^{2}+K_{2}\sum_{k=1}^{N}L\left(x_{k},x_{k-1},u_{k},u_{k-1}\right)\;s.t.\;\;\;\mathrm{for}\;k=0,1,2,\mathrm{IN}-1\;\;\;\;\;\bar{g}\left(x_{k},u_{k},x_{k+1},u_{k+1},\lambda_{k},\lambda_{k+1},{\bar{\lambda }}_{k},{\bar{\delta }}_{k}\right)\;=\;{\dot{\bar{x}}}_{t_{c,k}}\;-\left[\begin{array}{c} v_{t_{c,k}}+J\left(q_{t_{c,k}}\right){\bar{\delta }}_{k}\\ \bar{f}\left(x_{t_{c,k}},u_{t_{c,k}},{\bar{\lambda }}_{k}\right) \end{array}\;\right]=0,\;\;m\left(x_{k},u_{k}\right)\leq 0,\;\mathrm{kinematic}\;\mathrm{manifold}\;\mathrm{constraints}\{\begin{array}{c} \varphi_{j}\left(q_{k}\right)=0,\\ {\dot{\varphi }}_{j}\left(q\right)=\phi \left(q_{k},v_{k}\right)=0,\\ {\ddot{\varphi }}_{j}\left(q\right)=\psi \left(q_{k},v_{k},{\dot{v}}_{k}\right)=0, \end{array}\;j\in \;\left\{{\mathrm{mode}}_{0},{\;\mathrm{mode}}_{1},{\;\mathrm{mode}}_{2}\right\}\;\mathrm{virtual}\;\mathrm{knot}\;\mathrm{constraints}=\{\begin{array}{c} J{\left(q_{f_{i}}\right)}^{T}{\mathrm{Impluse}}_{i}=K_{O}M\left(q_{f_{i}}\right)\left(v_{0_{i+1}}-v_{f_{i+1}}\right)\\ q_{f_{i}}=q_{0_{i+1}} \end{array}\;i\in \;0\;\mathrm{or}\;1,\;\mathrm{footholds}\;\mathrm{constraint}=\{\begin{array}{c} \begin{array}{c} x_{1}=\left(0.5+{\mathrm{slack}}_{0}\right)(x_{0}+x_{2})\\ a\leq {\mathrm{slack}}_{0}\leq b \end{array}\\ a=-0.05\\ b=0.05 \end{array},\;\mathrm{periodic}\;\mathrm{symmetry}\;\mathrm{constraint}=\{\begin{array}{c} \overset{n-1}{\underset{t=0}{\sum }}h_{t}=\overset{m-1}{\underset{t=0}{\sum }}h_{t}+\overset{p-1}{\underset{t=0}{\sum }}h_{t}=H\\ x_{\mathrm{lf}\_m_{0}}=x_{\mathrm{rf}\_m_{1}}\\ x_{{\mathrm{rf}}_{m_{0}}}=x_{{\mathrm{lf}}_{m_{1}}}\\ \mathrm{Initial}\;\mathrm{state}=\mathrm{Final}\;\mathrm{state} \end{array}\;\mathrm{Contact}\;\mathrm{friction}\;\mathrm{constraints},\;\mathrm{Other}\;\mathrm{constraints}.$$

z is still the set of all of the decision variables, such as the time step, joint states, nputs, contact forces, correction slack variables, footholds, etc. $x_{k}{\;\mathrm{and}\;u}_{k}$ are the robot states and the joint inputs at the kth knot. $x_{0}$ and $x_{N}$ represent the initial state and final state. $h_{m}$ is the time step between two adjacent knots, and $t_{c,k}$ is the time index of the collocation point between k and k+1. ${\mathrm{Impluse}}_{i}$ is a slack variable of the ground reaction forces at the ith contact. ${\mathrm{slack}}_{0}$ is the slack variable of the foothold coordinate, and j is the mode index. The collocation constraints and kinematic manifold constraints are preserved from DIRCON. The virtual knot constraints are used for multi-phase planning, which is the basis of three-phase planning. The  footholds constraint and periodic symmetry constraint together comprise the approximate linear slacked symmetry constraint, which can be used in other optimization-based biped gait symmetry plans. m(z) represents other constraints, such as joint limits, velocity limits, and input limits during the whole trajectory. Contact friction constraints means that the contact point should not slip. In this work, our contact model is a two-point contact model wherein the midpoint of the front and rear edges of the supporting foot should maintain contact with the ground. Other constraints include some gait pattern constraints, such as the max height of the swing foot, walking speed, joint limits, the boundary of COM, etc., and, in this paper, we require that the highest position of the swing foot does not exceed 10 cm. Due to the cost function, we assigned different weights to the cost function for different purposes. The first two terms are the initial state cost and the final state cost, which are lightly weighted and only used to induce a standing posture for the initial state and final state. The third term is the highly weighted cost of energy consumption. The next term is used for the foothold slack variable described in Section 3.1, and here we direct this slack to zero. The last term is the smoothness cost of the joint trajectories, which is the square of the difference between the robot states and the inputs of the adjacent knots. $K_{0}$, $K_{1}$, and $K_{2}$ represent the weight of energy consumption, location selection, and trajectory smoothness. In our implementation, $K_{0}$ is set to 6.5, $K_{1}$ is set to 3.0, and $K_{2}$ equals 1.5. After solving the optimization, the joint position trajectories and input trajectories are restored by using the Hermite spline and first-order curves, respectively. Then a sampling at 100 Hz is conducted to obtain the target trajectories for tracking.

## 4. Robot Device Design and Implementation

To verify the planner, we designed a new biped robot with 10 active Dofs. Currently, our robot is designed to perform dynamic walking on flat terrain and does not use passive elastic components, such as leaf springs and torsion springs. In order to reduce the weight, the main materials used were an aluminum alloy and high-strength carbon fiber, which were used for joint connection and for the trunk, respectively. The structural design is shown in Figure 2, showing that the size of the robot is about 1.2 m×0.405 m×0.275 m. Overall, 13 links are present in the robot, including one baselink, two hip roll links, two hip yaw links, two hip pitch links, two thighs, two shanks, and two feet. Two passive fixed joints are indicated with red circles between the hip roll links and baselink. Ten revolute joints are numbered in green. The current robot was designed to mimic the lower body of a 1.85-m-tall human male. According to the morphology data in [26], the waist height accounts for 65% of the total height. Therefore, the robot’s height was designed to be lower than 1.23 m, and it has an actual height of 1.211 m. The range of each revolute joint is revealed in Table 1, and the joint reference frame is consistent with the walking frame shown in Figure 2. Each active joint has a DC servo motor controlled via CAN. The rated torque of the ankle is 38.7 Nm and 13.0 Nm for others. The overall mass is 18.532 Kg, and the mass of each link is shown in Table 2. The CoM is located at the mid-top between the two legs. The sub-graph of Figure 2 shows two kinds of feet; in this paper, a flat foot was used when validating the planner; the curved ebonite foot will be used to perform 3D walking in future work. The kinematic performance was exhibited through a gait stepping in the air, which is recorded in Part I of the supporting video. The model file (URDF) can be found in the Appendix A.

For the control system, a PICO-ATX-i7 with a Xenomai real-time kernel is installed as the main controller, and a multi-tread program is used as the master control software. All of the non-ankle motors are driven by an MIT-style driver board with a CAN interface [27], and the output torque can be calculated by following Equation (8):(8)$$\mathrm{Output}\;\mathrm{Torques}\;=\;P\;*∆\mathrm{pos}+D\;*∆\mathrm{vel}+f_{d}$$
where P is the position gain of the difference between the current joint position and the target position. D is the velocity gain of the difference between the current joint velocity and the target velocity. $f_{d}$ represents the compensation torque of friction, damping, and control performance. The communication of the whole control system is established via a CAN network built using four connection cubes, as shown in Figure 3. A signal adapter of USB-CAN is engaged to transfer the signal from the main controller to the CAN network, and a remote power switch and voltage conversion modules are inserted between the batteries, main controller, and one-chip computer. After testing, the control frequency can achieve around 315 Hz. In future work, more onboard devices and feedback data will be inserted into the control loop; therefore, the control frequency of the real-time system should be lower. Moreover, our simulation frequency in CoppeliaSim is set to 100 Hz, and to be consistent with the simulation, we set the control frequency to 100 Hz.

Sensors are another key component of the control system. Four force sensors are implanted in each foot, and a high-precision IMU is located near CoM, which is the main method to obtain the base attitude data. A 12-bit encoder is installed inside each motor, which returns information relating to the joint position, velocity, and current. All of the data and signals are transmitted through barrier wires inside the trunk. The onboard equipment and its locations are summarized in Table 3 and Figure 3.

## 5. Experiments

In this section, the proposed planner and the real robot device are verified through both dynamic simulation and physical comparison walking experiments.

### 5.1. Implementation Details of Simulation and Physical Experiments

The processor used to solve the optimization was “Core i7 9750H” which was produced by Intel in 2019, and the software environment used was Ubuntu 18.04, C++ and CMake 3.10 were used as the implementation language and compilation tools, respectively. We utilized a robot toolbox called “Drake” [28] to implement our planner and an efficient nonlinear program solver, “SNOPT”, which was designed by Stanford Business Software Inc and was integrated in Drake tosolve the optimization. Normalized random trajectories, which are modulated by the parameters of the robot’s structural design, were introduced as the initial estimate. As for the other priors, the max joint torques should not exceed 13.0 Nm, the velocities should also be smaller than 6.28 rad/s, and the GRF is initialized to the gravity of the robot. Due to the details of the simulation, Coppeliasim 4.2.0 [29] was used as the simulation platform, and the dynamic engine, “Newton”, implemented the physics. A control program, which sends the target trajectories and triggers the simulation step-by-step, was completed through the use of the Remote API of CoppeliaSim. The control mode of each active joint is set to the PD position control, and the friction coefficient of the contact face is set to 0.6. The time step of the dynamic engine is also set to 10 ms, matching the frequency of the sampling frequency of the target trajectory. It is worth noting that the dynamic model in the planner is planar, and there exist three floating coordinates, namely ‘X’, ’Z’, and ‘Pitch.’ Correspondingly, the simulation is also 2D. Finally, the gait is well-tracked during the simulation, enabling the robot to walk stably across a sagittal plane. As for the physical experiment, a treadmill was used as the flat terrain, and the coronal motion of the robot was also restricted by a limiter. The control program was reimplemented in a real-time manner on the main onboard controller, maintaining a frequency of 100 Hz.

### 5.2. Dynamic Simulation

To verify the planner and the structural design before executing an experiment on a real robot, we performed a planar dynamic simulation in CoppeliaSim 4.2.0. The complete gait, which contains three modes of 38 knots and two virtual knots, was generated from our planner. The contact points between the supporting foot and the terrain are located at the midpoints of the front and rear edges, and this contact resulted in multiple kinematic manifold constraints. The virtual knot and symmetrical constraints were enabled, as shown in Equation (7), and the boundaries of the footholds were set to −0.05 and 0.05, respectively. Moreover, the gait speed should match the velocity adjustment accuracy of the treadmill. Therefore, the total duration, H, is explicitly set to 0.72 s, and the step length of the complete gait is 0.2 m, i.e., a gait speed of 1.0 Km/h. m(z) is set according to the joint ranges shown in Table 1, and the friction constraints are maintained similarly to DIRCON. The solving time ranges from sixty-five seconds to five minutes. The generated position and input trajectories are depicted in Figure 4. From the joint trajectories, we find that the timing and amplitude between the left leg and right leg joints are symmetrical w.r.t. different supporting phases, and the velocities are also approximately the same. The input trajectories illustrate that the accelerations also maintain the same symmetric pattern, and the duration of each supporting phase is nearly identical. Therefore, our planner provides a dynamic feasible walking gait with a symmetrical pattern. Moreover, given the smooth cost function, the generated trajectories are relatively smooth, benefitting trajectory tracking. Finally, the trajectories were tracked using PD position control, and the five key frames are recorded in Figure 5 and the tracking performance is shown in Figure 6. In Figure 5, the first, third, and fourth frames represent transition events between the LSP and RSP, i.e., the right supporting foot taking off, the right swing foot touching down, and the left supporting foot taking off. The second and fifth frames are the two typical robot states where the swing foot reaches the highest position. After the simulation, our planner and the robot’s design have been preliminarily verified.

In our previous work [18], we designed a biped gait planner with an emphasis on walking on non-flat terrain. Energy efficiency and morphology characteristics were not considered during planning. One finds that the robot walks slightly stiffly, and the joints behave differently in different supporting phases. In our current work, these essentials can be introduced to the proposed planner. Compared with the flat terrain gait planner in [18], we find that the proposed planner is solved faster, and the max joint input is also smaller for the same gait parameters. In other words, the proposed planner achieves a more efficient and natural gait pattern. Table 4 compares the flat terrain gait of the two planners.

### 5.3. Physical Comparison Walking Experiments

After the simulation, we conducted physical walking experiments on our real robot to study the proposed planner and the original planner based on DIRCON. As a counterpart, the original planner only maintains some necessary constraints, such as collocation constraints, kinematic manifold constraints, virtual knot constraints, and contact friction constraints, and the cost function keeps the form without a slack term. The walking speeds are maintained at 1.0 Km/h as per the simulation. Due to the characteristic of planar walking, the joints are divided into main joints and non-main joints. The main joints include the pitch joints, knee joints, and ankle joints, which execute the motion of walking in a sagittal plane, and the rest of the joints are non-main-joints, which are required to maintain zero to limit the leg behavior in the coronal plane. The two gaits were generated offline and transferred to a motor driver through the real-time multi-thread program on the main controller at 100 Hz, and a PD position control with forward torque compensation was implemented to track the trajectory. The position and torque (current) data are returned during the same cycle. Figure 7 shows the feedback torque of each motor in two steps, and the ankles are delivered separately because of an external gearbox (1:20) that is engaged, and the unit of torque feedback is represented in Amps. Due to the full-order dynamics used in the planner, both the non-symmetrical gait and symmetrical gait can enable the robot to walk on the treadmill. Figure 8 presents the target trajectories and the tracking performance of each joint in two steps. Figure 9 and Figure 10 record five key frames of one complete gait of non-symmetrical walking and symmetrical walking, respectively. The first, third, and fourth frames describe the transition events, and the second and fourth frames are two typical robot states where the swing feet are swinging forward. The white rectangles represent the step length of two walking gaits, and we can clearly observe that the gait planned by the proposed planner is symmetrical, which enables the robot to walk naturally, and the gait planned by the original planner cannot. The experiment has been recorded in Part II and Part IV of the supporting video, and in order to avoid accidental deviation, other non-symmetrical walking experiments were also conducted and repeated, and the experiments were recorded in Part III, which shows similar non-symmetrical locomotion and recorded similar results.

### 5.4. Analysis

In the comparative walking experiments, we conducted two planar walking experiments between the proposed planner and a counterpart planner on our newly designed robot, and an analysis of the morphological characteristics, joint trajectories, and energy efficiency was also performed.

The complete gait of non-symmetrical walking, which is shown in Figure 9, starts from the taking off of the left foot in the RSP and ends at touching down in the LSP. The white dashed rectangles demonstrate that the step length of the RSP is much longer than the LSP, i.e., the robot is limping on the treadmill, which can be observed clearly in the supporting video. Relatively, the gait generated by our proposed planner shows a more natural pattern in Figure 10, where the step length of the LSP and RSP are almost identical. It means that the approximately linear foothold constraint works well in three-phase planning. On the other hand, due to the duration constraint, the left swing foot and right swing foot swing at almost the same speed in the LSP and RSP so that we can observe the robot stepping naturally.

From the joint position trajectories in Figure 8, we can also find that the amplitudes of the left and right legs of the symmetrical gait are equal, and the shapes and timing are also similar. However, a significant difference can be observed in the non-symmetrical gait. This means that the state exchange constraint during the transitions effectively introduces the symmetry requirement at the joint level. The same phenomenon can also be seen in the input trajectories in Figure 7, showing that the input trajectories of the symmetrical gait changed regularly, and the unsymmetrical gait behaved desultorily, especially for the pitch joints and knee joints. The peak torque of the symmetrical gait is observed at the pitch joints of 8.4 Nm, and the non-symmetrical gait takes 10.2 Nm without consideration of the ankles. Regularity and small peak torque imply that the gait is easier to implement and track by the controller, i.e., the proposed planner can provide more realizable joint trajectories with specific patterns for walking on flat terrain.

Due to the energy efficiency, we calculated the Specific Resistance (SR) [30] of the two gaits through the integration of the input feedback, and the principle is the same as the cost of transport (COT). The calculation equation is shown in Equation (9), where E is the energy consumption, M is the total mass of the robot, and d is the forward distance. The SR of the symmetrical gait achieves 0.772, and the unsymmetrical gait is 1.299. The comparison demonstrates that the proposed planner can indeed provide a more efficient gait pattern than the original planner. The quantity of SR is also relatively small compared to others [7]. It is worth noting that it is still higher than we expected. We think that the main reasons include the stiff contact pattern, not using gait parameter optimization, and not having a passive spring.
(9)$$\mathrm{SR}\;=\frac{E}{\mathrm{Mgd}}\;$$

## 6. Conclusions

In this paper, we propose a DIRCON-based full-order biped gait planner wherein the symmetry requirement, energy efficiency, and trajectory smoothness can all be involved at the joint level. A symmetrically efficient gait can be easily generated by the proposed planner through the use of three-phase planning. Furthermore, a newly designed biped robot is also proposed. Dynamic simulations and physical comparison walking experiments were both conducted to verify the proposed planner and robot design, and the experiment results demonstrate that the generated gait achieves good symmetry and high energy efficiency. The solution in this work preliminarily realizes our intention of developing a new and efficient bipedal robot. However, limitations still exist. The solving time takes too long to implement in an online manner, and energy dissipation is severe due to a stiff contact pattern. Gait parameter optimization is not considered during planning, leading to additional energy loss. Some onboard equipment performance is not enough for 3D dynamic walking, such as communication adapters. In the future, we will simplify the dynamic model to accelerate the planner, and an online control system will be developed for its use in relation to 3D walking. The robot device will also be upgraded and use some spring components to store energy.

## Figures and Tables

**Figure 1 micromachines-14-00417-f001:**
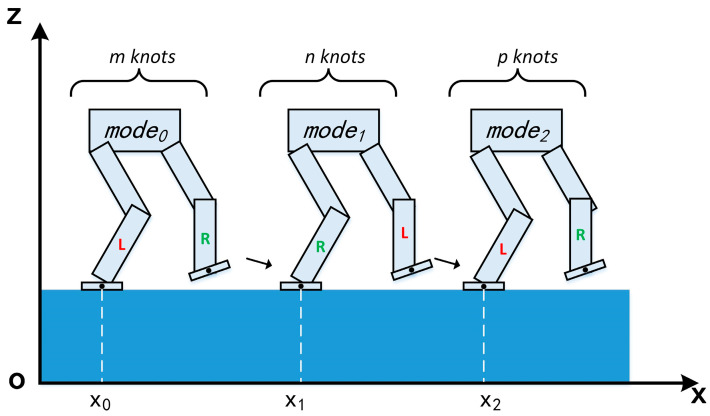
Decision variables of three footholds in one single gait planning.

**Figure 2 micromachines-14-00417-f002:**
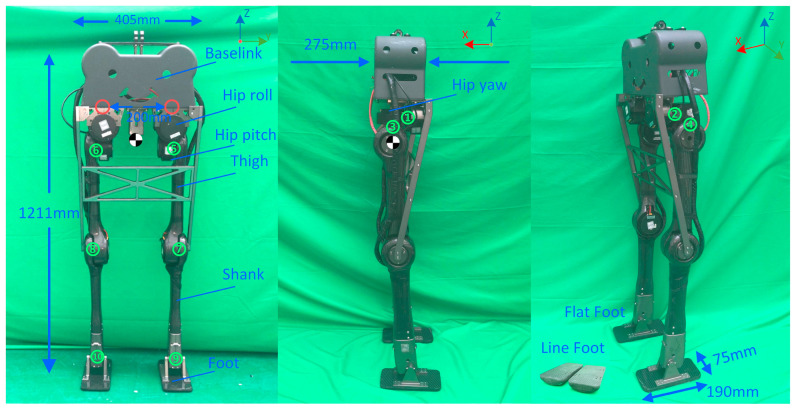
Composition and size of the robot. The left diagram shows joint distribution that fixed joints marked in red and actuated joints numbered in green. The middle diagram shows that the width of the robot is 275 mm. Two types of feet are shown in the right diagram and the length of the feet used in this paper is 190 mm and the width is 75 mm.

**Figure 3 micromachines-14-00417-f003:**
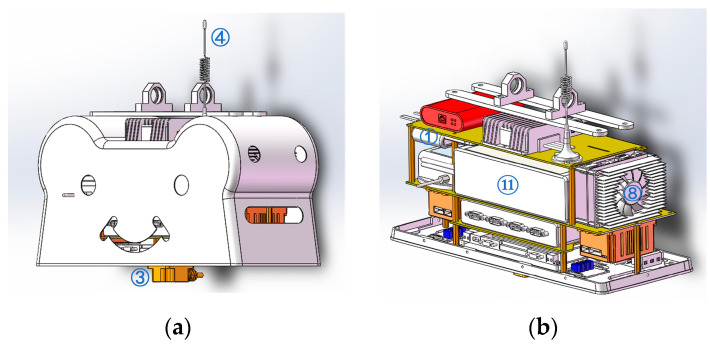
The layout of on-board equipment. The main equipment is numbered according to Table 3. (**a**) Base link appearance. (**b**) Inside oblique drawing (**left**). (**c**) Inside oblique drawing (**right**). (**d**) Inside oblique drawing.

**Figure 4 micromachines-14-00417-f004:**
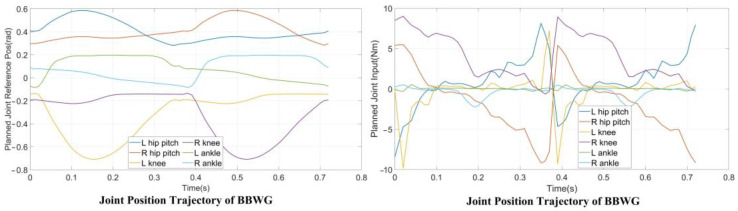
Planned symmetrical joint-level trajectories. In the left diagram, significant symmetry can be found and all the trajectories are within the kinematic limits. Reference joint inputs are shown in the right which the max torque is 9.8 Nm.

**Figure 5 micromachines-14-00417-f005:**
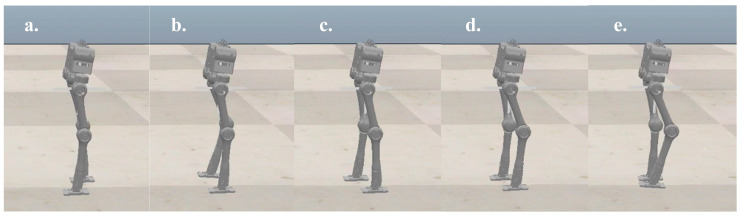
Five key frames of dynamic simulation. (**a**) The right supporting foot is taking off from the ground. (**b**) Right foot is swinging forward. (**c**) Right swing foot is touching down (**d**) Left foot is taking off from the ground. (**e**) The left foot is swinging forward.

**Figure 6 micromachines-14-00417-f006:**
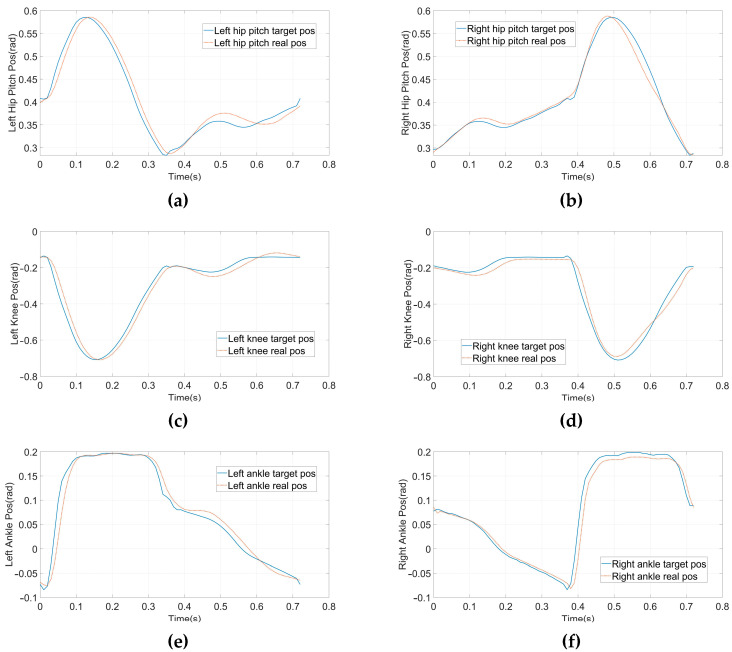
Generated position trajectories and tracking performance in CoppeliaSim. (**a**) Left Hip Pitch position. (**b**) Left Hip Pitch position. (**c**) Left Knee position. (**d**) Right Knee position. (**e**) Left Ankle position. (**f**) Right Ankle position.

**Figure 7 micromachines-14-00417-f007:**
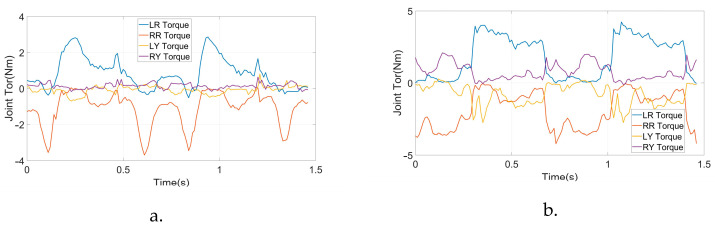
Feedback input trajectories of each active joint. (**a**) Roll and Yaw input of symmetric gait. (**b**) Roll and Yaw input of non-symmetric gait. (**c**) Pitch and Knee input of symmetric gait. (**d**) Pitch and Knee input of non-symmetric gait. (**e**) Ankle input of symmetric gait. (**f**) Ankle input of non-symmetric gait.

**Figure 8 micromachines-14-00417-f008:**
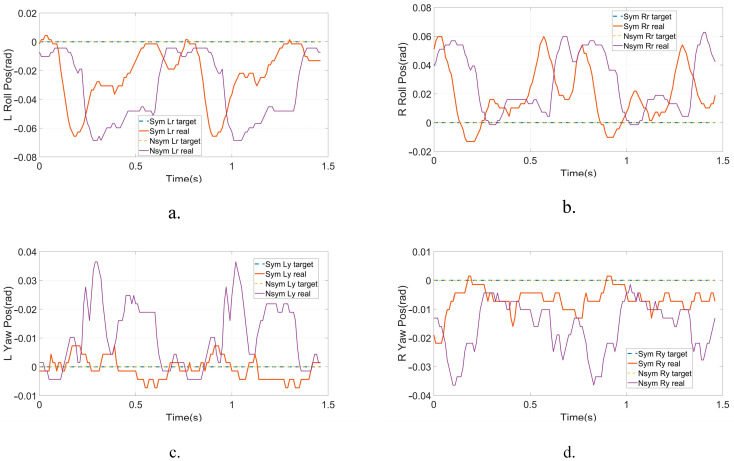
The positions and tracking performance of each active joint for two steps. (**a**) Left Hip Roll tracking performance. (**b**) Right Hip Roll tracking performance. (**c**) Left Hip Roll tracking performance. (**d**) Right Hip Roll tracking performance. (**e**) Left Hip Pitch tracking performance. (**f**) Right Hip Pitch tracking performance. (**g**) Left Knee tracking performance. (**h**) Left Knee tracking performance.(**i**) Left Ankle tracking performance. (**j**) Right Ankle tracking performance.

**Figure 9 micromachines-14-00417-f009:**
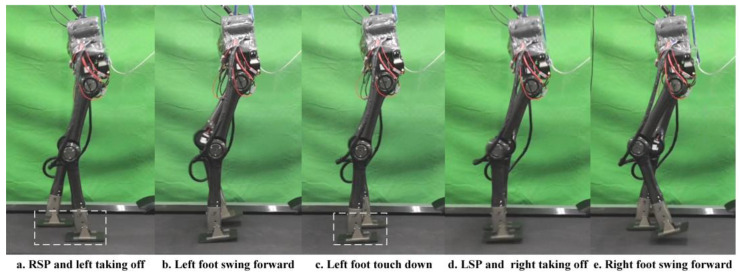
Five key frames of non-symmetrical walking experiment.

**Figure 10 micromachines-14-00417-f010:**
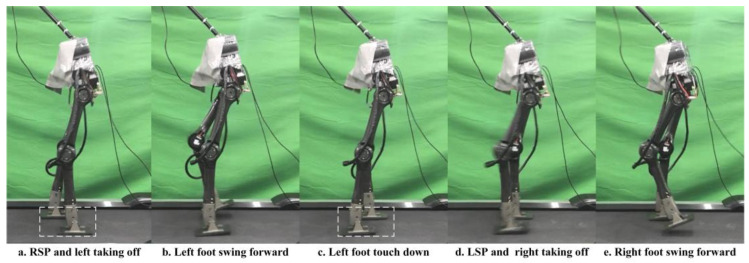
Five key frames of symmetrical walking experiment.

**Table 1 micromachines-14-00417-t001:** Joint Range.

Joints	Angle (deg)	Joints	Angle (deg)
Baselink-Hip roll	Fixed	Hip Pitch Joint(⑤,⑥)	[−47,+47]
Hip Roll Joint(①,②)	[−15,+95]	Knee Joint(⑦,⑧)	[−63,+63]
Hip Yaw Joint(③,④)	[−45,+45]	Ankle Joint(⑨,⑩)	[−85,+85]

**Table 2 micromachines-14-00417-t002:** Mass Distribution.

Components	Mass (kg)	Components	Mass (kg)
Baselink	7.624	Thigh	1.116
Hip roll	0.922	Shank	1.597
Hip yaw	0.776	Foot	0.300
Hip pitch	0.743	Total	18.532

**Table 3 micromachines-14-00417-t003:** On-board Equipment.

Number	Device	Number	Device
1	USB-HUB	9	On-chip computer of contact sensor
2	The signal adapter of contact sensors	10	Voltage adapter of One-chip computer
3	IMU and Attitude sensor	11	Main battery
4	The antenna of remote power switch	12	Auxiliary power supply
5	Remote power switch	13	The driver of the ankle motor
6	Cubes of CAN network and power	14	SSD of the main controller
7	Voltage adapter of the main controller	15	USB-CAN-4EU
8	Main on-board controller	16	CAN Analyzer

**Table 4 micromachines-14-00417-t004:** c between two planners.

	Previous Flat Terrain Gait Planner	Proposed Symmetry Gait Planner
Solving time	120 s to 3 min (average 156 s)	65 s to 5 min (average 138 s)
Morphology	Slightly stiff	More nature
Max joint input	11.5 Nm	9.8 Nm
Symmetry	No	Yes

## Data Availability

Not applicable.

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
