# Peer review of "Symmetrical Efficient Gait Planning Based on Constrained Direct Collocation"

_micromachines, 2023, doi:10.3390/mi14020417_

Round 1

Reviewer 1 Report

Review of the manuscript by Boyang Chen et al.

Symmetrical Efficient Gait Planning Based On Constrained Direct Collocation

Submitted to Micromachines-2127582

GENERAL COMMENTS

Micromachines is an international, peer-reviewed, open access journal, which provides an advanced forum for studies on micro/nano-scaled structures, materials, devices and systems. It is known that this journal seeks and encourages submissions on significant and original works related to all aspects of micro/nano-scaled structures, materials, devices, systems as well as related micro- and nanotechnology from fundamental research to applications. It seems that this manuscript is out of scope, and it is suggested that this manuscript could be submitted to another journal ‘Symmetry’. This suggested journal covers Theories and Applications related to symmetry/asymmetry phenomena in all scientific studies. It has five broad subjects comprising the many symmetry Sections.

Author Response

Dear,

We are very glad to receive your comment of the manuscript and thank you very much for your work.

We have carefully considered your comment and checked the scope and aims, issues and current topics of Micromachines. Please see our explanation in the attachment. Hope it can clear the doubts.

Hope you are well.

Best Regards,

The authors

Reviewer 2 Report

Authors focused how to synthesize an energy efficient symmetrical walking gaits so authors proposed DIRCON based full-order biped gait planner. However, there are several English grammar mistakes in entire manuscript. Thus, authors need to check Englsih with native Englsi colleagues or professional English services. In addition to English, there are some suggestive comments as below before publication.

1. Please correct mass(CoM) to mass (CoM). Please check others in entire manuscript.

2. Reference format is wrong. Please correct [4][8][9] to [4,8,9].

3. Please change Eq. to Equation.

4. In Line 265, Fig 2 to Figure 2 need to be corrected.

5. Figure 4 labels size are too small. They are hard to be recognized.

6. Authors mentioned they are symmetical patterns in Figure 4 so please indicate where the patterns are symmetical.

7. Authors had better compare the previous work with proposed work so please provide comparison data in Table.

8. Authors must use abbreviated journal names in the reference section.

9. Authrs need to provide city, date, and country information of the conference papers.

10. Authors mentioned in the future work about 3D walking. How authors could proceed that ?

11. Authors need to mention the limitation of the proposed work.

12. Please check Table format according to MDPI policy.

13. Authors mentioned that there are 5 key frame in Figure 5. How to determine that ?

14. Authors mentioned that Specific Resistance (SR). How about the equations from the reference ?

15. Please correct Symmetrical Efficient gait planning to Symmetrical Efficient Gait Planning in Line 157.

Author Response

Dear,

We are very glad to receive your comments of the manuscript and thank you very much for your meticulous work.

We have carefully considered your comments and revised the manuscript. Please see the explanations and supplements in the attachment, which are also be done in the manuscript.

Hope you are well.

Best Regards,

The authors

Reviewer 3 Report

This paper reports a Symmetrical Efficient Gait Planning Based on Constrained Direct Collocation. The paper is interesting, and the proposed design could be potentially used in modern biped robots. I have the following comments/concerns about the submitted manuscript that could improve its overall quality and readability.

1.       The authors need to provide the main contribution and novelty of their proposed technique over the already reported in the literature. I suggest the authors include a table of comparison for this purpose.

2.       Why the authors have chosen to use CoppeliaSim as the software?

3.       What is the reason for choosing a seven-link XOZ planar biped robot?

4.       Why LSP and RSP are considered equal for joint states exchanges?

5.       The authors should explain the reason for using weighted cost functions for Symmetrical Efficient Gait Planning.

6.       What criteria were followed to decide the height of the robot to be 1.2 meters?

7.       I suggest that the component “feedewardtorque” in equation 8 should be replaced with some symbol.

8.       What was the reason for using the main controller at a frequency of 100Hz?

9.       There are many spelling mistakes throughout the manuscript. On page 1 line 5 “filed” should be replaced by field. On page 3 line 111 the word “showed” should be replaced with shown. On page 5 line 197, the word “and” is repeated. Also “the mass of each links” should be the mass of each link. I suggest that the authors get their manuscript proofread by a native speaker.

1.   Figure 4 is not very clear in the manuscript.

1.   On page 4, line 139, the sentence is not complete.

1.   There are many typos in the manuscript that need to be fixed.

Author Response

Dear,

We are very glad to receive your comments of the manuscript and thanks very much for your meticulous work. Thank you!

We have carefully considered your comments and revised the manuscript. Please see the explanations and supplements of your comments in the attachment, which are also be done in the manuscript.

Hope you are well.

Best Regards,

The authors

Round 2

Reviewer 1 Report

My first decision is “Reject”, and now I think this manuscript is still is out of scope. Hence, the final decision is made by the Editor, and no further comment is provided here. More reviewers might be invited on this manuscript.

Author Response

Dear,

We are very glad to receive your comment of the manuscript and thank you very much for your work.

We fully understand your concerns. We also have carefully checked the scope and issues, current topics and relevant papers, and we believe that our work can attract a lot of attention and extend to discussion.  

Thanks again for your work and Hope you are well.

Best Regards,

The authors

Reviewer 3 Report

Thanks for updating the manuscript based on my comments. However, I still believe that extensive English language editing is needed.  

Author Response

Dear,

We are very glad to receive your comment of the manuscript and thank you very much for your work.

We have used the MDPI's English editing service and carefully revised the manuscript. The attachement is the certificate and changes was marked in red.  Thanks for your work and Hope you are well.

Best Regards,

The authors
